# Periprosthetic Joint Infection Prophylaxis in the Elderly after Hip Hemiarthroplasty in Proximal Femur Fractures: Insights and Challenges

**DOI:** 10.3390/antibiotics10040429

**Published:** 2021-04-13

**Authors:** Dolors Rodríguez-Pardo, Laura Escolà-Vergé, Júlia Sellarès-Nadal, Pablo S. Corona, Benito Almirante, Carles Pigrau

**Affiliations:** 1Infectious Diseases Department, Vall d’Hebron Hospital Universitari, Vall d’Hebron Barcelona Hospital Campus, Passeig Vall d’Hebron 119-129, 08035 Barcelona, Spain; lauraescola@gmail.com (L.E.-V.); juliasellares@gmail.com (J.S.-N.); balmiran@vhebron.net (B.A.); cpigrau@vhebron.net (C.P.); 2Spanish Network for Research in Infectious Diseases (REIPI RD16/0016/0003), Instituto de Salud Carlos III, 28029 Madrid, Spain; pcorona@vhebron.net; 3Study Group on Osteoarticular Infections of the Spanish Society of Clinical Microbiology and Infectious Diseases (GEIO-SEIMC), 28003 Madrid, Spain; 4Medicina Interna, Universitat Autònoma de Barcelona, 08193 Bellaterra, Spain; 5Septic and Reconstructive Surgery Unit (UCSO), Orthopaedic Surgery Department, Vall d’Hebron Hospital Universitari, Vall d’Hebron Barcelona Hospital Campus, Passeig Vall d’Hebron 119-129, 08035 Barcelona, Spain

**Keywords:** hip hemiarthroplasty, proximal femur fracture, antibiotic prophylaxis, periprosthetic joint infection, decolonization

## Abstract

We review antibiotic and other prophylactic measures to prevent periprosthetic joint infection (PJI) after hip hemiarthroplasty (HHA) surgery in proximal femoral fractures (PFFs). In the absence of specific guidelines, those applied to these individuals are general prophylaxis guidelines. Cefazolin is the most widely used agent and is replaced by clindamycin or a glycopeptide in beta-lactam allergies. A personalized antibiotic scheme may be considered when colonization by a multidrug-resistant microorganism (MDRO) is suspected. Particularly in methicillin-resistant *Staphylococcus aureus* (MRSA) colonization or a high prevalence of MRSA-caused PJIs a glycopeptide with cefazolin is recommended. Strategies such as cutaneous decolonization of MDROs, mainly MRSA, or preoperative asymptomatic bacteriuria treatment have also been addressed with debatable results. Some areas of research are early detection protocols in MDRO colonizations by polymerase-chain-reaction (PCR), the use of alternative antimicrobial prophylaxis, and antibiotic-impregnated bone cement in HHA. Given that published evidence addressing PJI prophylactic strategies in PFFs requiring HHA is scarce, PJIs can be reduced by combining different prevention strategies after identifying individuals who will benefit from personalized prophylaxis.

## 1. Introduction

Antimicrobial prophylaxis (AP) is crucial in preventing surgical site infections (SSIs) after orthopedic surgery, with a reduction by up to 81% in the relative risk of infection and 8% in the absolute risk [1]. In proximal femoral fractures (PFFs) with internal fixation, two metanalyses showed that AP reduced the incidence of SSIs compared to either non-prophylaxis or placebo [2,3]. The standard care AP in orthopedic surgery has traditionally been first-generation cephalosporins. This is due to their adequate spectrum for the general population, safety profile, and low price [1]. However, this approach may not always be adequate, particularly for institutionalized patients who have skin flora alterations and multidrug-resistant organisms (MDROs) colonization [4]. For this reason, some physicians may consider it more appropriate to provide them with individualized prophylaxis.

Other strategies such as cutaneous and nasal *Staphylococcus aureus* decolonization have proven to be effective in reducing early SSIs in orthopedic surgery [5,6,7]. However, their implementation may not be easy, their effectiveness is sometimes controversial, particularly with a low incidence of *S. aureus* SSIs, and it has not been specifically addressed in PFF in the elderly.

We aim to undertake a critical appraisal on current periprosthetic joint infection (PJI) prophylaxis strategies in PFFs requiring HHA surgery and exploring future research areas.

## 2. Current Antibiotic Prophylaxis in Proximal Femur Fractures Requiring Hip Hemiarthroplasty

Single-dose or continuation for less than 24 h AP is recommended for hip fracture repair in procedures involving prosthetic replacement or internal fixation [1,8]. Surgical AP should be administered within 120 min before incision. However, it is recommended that when using short half-life beta-lactams (e.g., first-generation cephalosporin drugs), they be administered within 60 min [1,8]. Single-dose or regimens of <24 h duration antibiotics that ensure drug concentrations during surgery will be appropriate [1].

The antimicrobial agent most commonly used in orthopedic procedure prophylaxis [1,8] is cefazolin. Even though cefuroxime has been used in PFF surgery in the elderly [9], second and third-generation cephalosporins are not routinely recommended here due to adverse events (i.e., *Clostridioides difficile*-associated diarrhea) and potential to cause antibiotic resistance [1].

In the case of beta-lactam type 1 (immunoglobulin E (IgE)-mediated) allergy, methicillin-resistant *S. aureus* (MRSA) colonization or a high prevalence of nosocomial MRSA SSI, clindamycin, or a glycopeptide (vancomycin or teicoplanin) may be used [1,8,10]. Although cross-allergic reactions between penicillin and cephalosporins are uncommon, cephalosporins should not be used for surgical prophylaxis in patients with documented or presumed IgE-mediated penicillin allergy [1]. Vancomycin is less effective than cefazolin for preventing SSIs caused by methicillin-susceptible *S. aureus* (MSSA). However, it is recommended with cefazolin in non-allergic patients [11]. Likewise, the addition of teicoplanin to cefazolin in arthroplasty surgery reduced PJIs thanks to a decrease in Gram-positive bacterial infections [12].

When there is an increased risk of Gram-negative bacilli (GNB) SSIs (i.e., colonized or recently infected patients), published guidelines recommend glycopeptides added to (1) cefazolin or cefuroxime in the absence of beta-lactam allergies and (2) aztreonam, gentamicin, or single-dose fluoroquinolone if there are allergies [1,8]. Although studies are not specific on PFFs requiring HHA, these recommendations are followed in the absence of more specific ones.

## 3. Current Challenges to Optimize Antibiotic Prophylaxis in Proximal Femur Fractures Requiring Hip Hemiarthroplasty

Patients with PFFs undergoing HHA are usually elderly, frail, comorbid, recently hospitalized, or even institutionalized. Consequently, standard AP may not be as effective as expected and should be individualized according to local epidemiology and antimicrobial susceptibility patterns [8]. Hence, the usefulness of strategies such as MDROs decolonization or individualized AP should be considered.

Regarding skin decolonization, there is considerable experience in *S. aureus* [5,6,7] which is the first cause of acute PJIs after total joint and HHA. Thus, a cohort study including 19 hospitals in Spain [13] showed a total of 7.9% (95% CI: 6.8–9.1%) MRSA-caused PJIs. Decolonization with intranasal mupirocin prevents SSIs in orthopedic surgery in patients with documented *S. aureus* [1,5,6]. However, identifying and specifically treating colonized individuals is a costly and challenging process. It requires a complex structure that allows screening, obtaining the results, and performing five days of nasal decolonization treatment with mupirocin before surgery. These steps are complicated to coordinate and done on time since PFFs require emergency surgeries. In this context, universal decolonization is the suggested alternative despite the risk of developing resistance to mupirocin which has been considered low [14]. Other noteworthy approaches are universal preoperative nasal and skin decolonization with chlorhexidine bathing in addition to the alcohol-based nasal antiseptic application [15] or chlorhexidine washcloths and oral rinse and intranasal application of povidone-iodine solution the night before and the morning of scheduled surgery [7]. Both strategies reduced PJI rates, associated morbidity, and costs thus avoiding resistance to mupirocin. However, although preoperative chlorhexidine bathing is widely performed in real-world practice, nasal decolonization is not, which probably allows for improvement in this strategy’s outcomes.

In the Spanish study [13] mentioned above, a statistically significant rising linear trend was observed for those PJIs caused by aerobic GNB (25% in 2003–2004, 33.3% in 2011–2012; *p* = 0.024) globally and also by MDR-GNB (from 5.3% in 2003–2004 to 8.2% in 2011–2012; *p* = 0.032). We have also published our experience regarding PJIs in patients undergoing HHA secondary to PFFs [16]. Among a total cohort of 381 patients included between 2011 and 2013, PJIs were diagnosed in 21 (5.51%), with a significantly higher incidence of SSIs among chronic institutionalized vs. non-institutionalized (9.52% vs. 3.99%; *p* = 0.04). Remarkably, GNB were the principal pathogens involved (67% of all PJIs). These observations suggest that asymptomatic bacteriuria (ASB) and fecal and urinary incontinence, both common among the elderly, support skin colonization either before the surgery or in the immediate postoperative period. Different authors have addressed the relationship between ASB treatment and PJIs in hip or knee surgeries with controversial outcomes. Sousa et al. [17] found that ASB was an independent risk factor for PJIs, with no correlation found between previously isolated bacteria in the urine and PJIs, while Honkanen et al. [18] did not find any relation between preoperative bacteriuria and PJIs in primary hip or knee replacement surgeries. Besides, Cordero et al. [19] did not identify any PJIs of urinary origin in patients with ASB. All these studies included both patients undergoing total hip arthroplasties and HHA. However, when the analyzed cohort is reduced to geriatric patients undergoing PFF surgery, the results are more contentious, and some authors conclude that prevalent bacteriuria treatment decreases the risk of SSI [9,20]. We evaluated the clinical impact of preoperative ASB treatment with a single dose of 3 g of oral fosfomycin between 24 and 6 h before surgery vs. no treatment on the reduction of early-PJI after HHA in an open-label, multicenter randomized clinical trial (BARIFER CT, Eudra CT 2016-001108-47). A total of 594 patients were enrolled (mean age 84.3 years), of whom 152 (25%) had ASB (77 treated with fosfomycin and 75 not treated), and 442 (75%) controls did not have ASB. It was found that neither preoperative ASB nor its treatment are independent risk factors of early-PJI in HHA surgery. Therefore, we consider that routine screening and preoperative ASB treatment should not be recommended.

Some literature has been published regarding skin decolonization in patients with MDR-GNB (extended-spectrum beta-lactamase (ESBL) or carbapenemase-producing Enterobacterales). Huttner et al. [21] carried out a study in adults with an ESBL-producing Enterobacterales (ESBL-E) positive rectal swab. Fifty-eight patients were allocated 1:1 to either placebo or colistin sulfate (50 mg four times per day) or neomycin sulfate (250 mg four times per day) for up to ten days plus nitrofurantoin (100 mg three times a day) for up to five days in the presence of ESBL-E bacteriuria. It was observed that this regimen temporarily suppressed ESBL-E carriage but had no long-term effect after seven days. Given its limited efficacy and the time needed to implement the protocol, these strategies are not applicable in emergency surgeries such as HHAs for PFF. On that basis, some authors and guidelines support extending AP in high-risk of being colonized by MDR-GNB individuals [1,8].

A recently published experience by Cuchi et al. [22] evaluates the role of previous skin and urine colonization in the development of deep SSIs after PFF surgery. It failed to find a relationship between skin colonization, urine culture, and deep SSI.

As observed, it appeared that patients would not benefit from modifying current AP in HHA and ASB. Regarding cutaneous MDR-GNB colonization, there is no strong evidence but a small cohort of patients’ and experts’ opinions advising extending AP in patients at risk of MDR-GNB skin colonization. However, we would advise caution and act accordingly only when MDRO colonization is confirmed.

## 4. Future Scenarios to Optimize Prosthetic Joint Infection Prophylaxis in Proximal Femur Fractures Requiring Hip Hemiarthroplasty

Strategies for MDRO screening and decolonization need to be optimized. Recently, new molecular tools have been developed to rapidly identify MDROs in different clinical samples such as skin and rectal screening swabs by real-time polymerase-chain-reaction (PCR) and sequencing techniques. These can detect targeted genes within a few hours which is relevant in urgent surgeries such as HHA in PFFs. These highly sensitive and specific methods would rapidly determine not only MRSA/MSSA colonization [23] but also ESBL-E [24] and carbapenemase-producing Enterobacterales [25] carriers. Standardization of such techniques would allow individualized prophylaxis covering MDROs only in patients with proven colonization. As experience accumulates, it will be assessed whether this individualized prophylaxis reduces GNB infection risk and whether it is a cost-effective strategy.

Another field of study is the use of alternative antibiotic regimens. In our experience, trimethoprim/sulfamethoxazole in monotherapy (800/160 mg of cotrimoxazole during anesthesia induction followed by another dose after 12 h) is effective. It prevents MRSA infections among chronic institutionalized patients undergoing HHA [16]. Besides, it is easy to handle and shows good tolerability.

There is a lack of information about the need to address candiduria or candidal intertrigo when they are detected before HHA surgery. These are quite common in elderly individuals who need diapering because of incontinence. In our experience, 34 (79.1%) out of 43 patients analyzed with *Candida* PJIs had at least one risk factor for *Candida* infection (six had concomitant intertrigo, and four showed candiduria before surgery) [26]. These data suggest that treating candidal intertrigo before HHA surgery could prevent PJIs easily. In contrast, it is not obvious whether candiduria should be addressed once we have observed that treating bacteriuria has no impact on reducing early-PJIs after HHA surgery.

Finally, the role of antibiotic-impregnated cement in primary HHA surgery is controversial. Antibiotic-impregnated bone cement is used as a spacer or during reimplantation surgery to treat infected total hip arthroplasties. A recent meta-analysis concluded it reduces infection rates by approximately 50% [27]. It has also been reported that high-dose dual antibiotic-impregnated (vancomycin and gentamicin) bone cement decreases PJIs rates in hip fractures [28]. Their use has recently become widespread in Spanish hospitals. This was reviewed during our multicenter randomized trial which assessed the impact of a PJI prevention strategy in patients with a PFF requiring HHA surgery (BARIFER CT data, Eudra CT 2016-001108-47). It was observed that 65.46% of HHA implant cases were cemented with antibiotics (64% with single and 36% with dual antibiotics). Given that some of the participating sites used them without changes in their specific AP, we hypothesize that this could justify a reduction in early-PJI rates compared to those previously reported between 2011 and 2013 (up to 9.52% among institutionalized patients) [16] and also in hospitals in our area [29]. Therefore, we encourage the use of antimicrobial-impregnated bone cement, and we also consider it interesting to be standardized in high-risk patients.

One of the major limitations of the opinion we share here is that the highest strength of evidence cannot always support recommendations due to the scarcity of published studies. Although certain antibiotics and prophylactic strategies may be discouraged or supported, final approaches should be tailored to local epidemiology and the antimicrobial stewardship programs at each center. We suggest a targeted preventive strategy, given that a broad-spectrum antibiotic regimen (i.e., meropenem plus linezolid or daptomycin), although covering possible MDROs, may result in new resistances (i.e., carbapenemases expression in Enterobacterales) and invalidate its future use.

## 5. Conclusions

Cefazolin might not be adequate for elderly and fragile patients with recent hospitalizations or institutionalization. In this scenario, our recommendations are (1) to expand AP to address MRSA or MDR-GNB in colonized or recently infected patients with such microorganism, (2) to perform universal preoperative nasal and skin decolonization accordingly the night before and the morning of surgery limiting the use of mupirocin for MRSA colonized patients and (3) to use dual antibiotic-impregnated (vancomycin and gentamicin) bone cement in primary HHA surgery. Thus, PJIs can be reduced by combining all these strategies after identifying those patients who may benefit from using personalized SSIs prophylaxis.

## Data Availability

Data presented in this study regarding BARIFER CT, Eudra CT 2016-001108-47 are available upon request from the corresponding author. BARIFER randomized clinical trial manuscript has been recently accepted for publication in the European Journal of Clinical Microbiology & Infectious Diseases.

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
