# Peer review of "Periprosthetic Joint Infection Prophylaxis in the Elderly after Hip Hemiarthroplasty in Proximal Femur Fractures: Insights and Challenges"

_antibiotics, 2021, doi:10.3390/antibiotics10040429_

Round 1

Reviewer 1 Report

Dear Authors

The present study covers a very important and hot topic: antibiotic prophylaxis in proximal femoral fractures of the elderly.

In general the study is well written, sound and properly structured.

I suggest revising the manuscript by an English expert in medical literature to make it easier to read.

Otherwise, I only have some minor concerns

Title and abstract: fine

Introduction

Line 42-43: I would clearly state that cefalin is a good choice for general population other than elderly and fragile patients (not only because of price and safety, but also adequate spectrum for general population)

Line 70-76: it is important to emphasize that cephalosporins must be changed in type 1 allergy (and only in type 1 allergy).

Author Response

The present study covers a very important and hot topic: antibiotic prophylaxis in proximal femoral fractures of the elderly.

In general, the study is well written, sound and properly structured.’

We want to thank the reviewer for his positive comment.

I suggest revising the manuscript by an English expert in medical literature to make it easier to read.

Thank you for your suggestion. Following your recommendations, we enclose you a more formal manuscript revised by an English expert in medical literature (¨For assessment¨ and ¨For publication¨ versions).

‘Otherwise, I only have some minor concerns.

Title and abstract: fine

Introduction

Line 42-43: I would clearly state that cefazolin is a good choice for general population other than elderly and fragile patients (not only because of price and safety, but also adequate spectrum for general population).’

Following reviewer’s suggestion, it has been clearly stated that cefazolin is a good choice for general population other than elderly and fragile patients (pages 1-2, line 44-48, ¨For assessment¨ version).

‘Line 70-76: it is important to emphasize that cephalosporins must be changed in type 1 allergy (and only in type 1 allergy).’

We appreciate the reviewer suggestion. Therefore, we have incorporated the following: ¨Although cross-allergic reactions between penicillin and cephalosporins are uncommon, cephalosporins should not be used for surgical prophylaxis in patients with documented or presumed Ig E-mediated penicillin allergy¨ (page 2, lines 74-79, ¨For assessment¨ version)

Reviewer 2 Report

Specific comments: Line 77: what is the increased risk of gram-negative bacilli? Which conditions?

It is not a clear review and has no clear question or hypothesis.

Clear recommendations are needed.

Author Response

REVIEWER #2:

 Comments and Suggestions for Authors

‘Specific comments: Line 77: what is the increased risk of gram-negative bacilli? Which conditions?’

Following the reviewer suggestion, we have specified those situations that we consider determine a higher risk for GNBs infections: ¨patients known to be colonized or recently infected with GNBs¨ (page 2, lines 85-86, ¨For assessment¨ version),

‘It is not a clear review and has no clear question or hypothesis.’  

As stated by the reviewer, this work is not a systematic review but a¨ Perspective article¨, which is what was commissioned by the editorial committee. Our objective is to express our opinion, based on our experience and those published works that we consider relevant, to optimize surgical prophylaxis in patients with proximal femur fractures subject to hip hemiarthroplasty. As we specify in our work ¨We aim to undertake a critical appraisal on current periprosthetic joint infection (PJI) prophylaxis strategies in PFFs requiring HHA surgery and exploring future research areas¨ (page 2, lines 57-59 ¨For assessment¨ version).

‘Clear recommendations are needed.’

Following your advice, a total of three recommendations have been stated in the Conclusions section (page 5 lines 241-250 For assessment version).

1) to expand AP to address MRSA or MDR-GNB in colonized or recently infected patients with such microorganism,

2) to perform universal preoperative nasal and skin decolonization accordingly the night before and the morning of surgery, limiting the use of mupirocin for MRSA colonized patients and

3) to use dual antibiotic-impregnated (vancomycin and gentamicin) bone cement in primary HHA surgery

Reviewer 3 Report

In general the manuscript is a good. The topics is really hot. 

I recommend a few modification.

The English written style should be revised by a English native speaker and the language requires some reconsideration in order to remove grammar and spelling inaccuracies and to make the manuscript more formal.

Can you reexplain the paragraph: between row 162- 172 need to be rewritten.

Conclusion need to be short and clear. Please phrase 3 short conclusion.

Please recheck the References order.

Author Response

‘Comments and Suggestions for Authors

In general, the manuscript is a good. The topics is really hot. 

We want to thank the reviewer for his positive comment.

I recommend a few modifications.

The English written style should be revised by a English native speaker and the language requires some reconsideration in order to remove grammar and spelling inaccuracies and to make the manuscript more formal.’

Thank you for your suggestion. Please find enclosed a further revision of the manuscript revised by an English native speaker to remove inaccuracies and to make the manuscript more formal.The incorporated changes are visible in ¨For assessment¨ and ¨For publication¨ versions.

Can you reexplain the paragraph: between row 162- 172 need to be rewritten.

Following the reviewer’s suggestion this paragraph has been rewritten (page 4, lines 176-191 ¨For assessment¨ version).

Strategies for MDRO screening and decolonization need to be optimized. Recently, new molecular tools have been developed to rapidly identify MDROs in different clinical samples, such as skin and rectal screening swaps. by real-time polymerase-chain-reaction (PCR) and sequencing techniques. These can detect targeted genes within a few hours which is relevant in urgent surgeries such as HHA in PFFs. These highly sensitive and specific methods would rapidly determine not only MRSA/MSSA colonization [23] but also ESBL-E [24] and carbapenemase-producing Enterobacterales [25] carriers. Standardization of such techniques would allow individualized prophylaxis covering MDROs only in patients with proven colonization. As experience accumulates, it will be assessed whether this individualized prophylaxis reduces GNB infection risk and is a cost-effective strategy.

Conclusions need to be short and clear. Please phrase 3 short conclusions.

Following your suggestions 3 short recommendations have been stated in the Conclusions section (page 5 lines 241-250 ¨ For assessment¨ version) 

1) to expand AP to address MRSA or MDR-GNB in colonized or recently infected patients with such microorganism,

2) to perform universal preoperative nasal and skin decolonization accordingly the night before and the morning of surgery, limiting the use of mupirocin for MRSA colonized patients and

3) to use dual antibiotic-impregnated (vancomycin and gentamicin) bone cement in primary HHA surgery

‘Please recheck the References order.’

References order and stile has been rechecked.

Round 2

Reviewer 2 Report

The paper deals with an interesting question of antibiotic prophylaxis in hemiarthroplasty or total joint arthroplasty of proximal femoral fractures.

However, the review is a selected collection of articles with do not clearly support the opinion of the authors. The conclusions are based on hypotheses and opinions of the authors, not on a clear evidence.

Therefore, I cannot recommand to publish that paper.